
**Retrieval of $CO_2$, $CH_4$, CO and $N_2O$ using ground-based FTIR data and validation against satellite observations over the Shadnagar, India.**

Mahesh Pathakoti[1,3]*, Sreenivas Gaddamidi[1,2], Biswadip Gharai[1], Sesha Sai Mullapudi Venkata Rama[1], Rajan Kumar Sundaran[3], Wei Wang[4]

[1]Atmospheric Chemistry and Processes Studies Division (AC&PSD), Earth and Climate Science Area (ECSA), National Remote Sensing Centre (NRSC), Indian Space Research Organization (ISRO), Hyderabad-500037, India
[2]Department of Physics, Jawaharlal Nehru Technological University Hyderabad-500085, India
[3]Lab for Spatial Informatics, International Institute of Information Technology (IIIT), Hyderabad- 500032, India
[4]Key Laboratory of Environmental Optics and Technology, Anhui Institute of Optics and Fine Mechanics, Chinese Academy of Sciences, Hefei, 230031, China

*Corresponding author: mahi952@gmail.com

## Abstract

An improved column averaged concentration ($X$) of greenhouse gases (GHGs) namely $CO_2$, $CH_4$, CO and $N_2O$ are retrieved using ground-based Fourier Transform Infrared (FTIR; model IFS125M) Spectrometer data collected at Atmospheric Science Lab (ASL) of National Remote Sensing Centre (NRSC), Shadnagar, India during 2016 period in clear sky days. Indium Antimonide (InSb) detector and Calcium Fluoride ($CaF_2$) beam splitter in combination with the spectral range between1800 $cm^{-1}$ to 11000 $cm^{-1}$ (5.50 μm to 0.90 μm) with 0.01 $cm^{-1}$ spectral resolution ($\Delta v$) are set for the present study. Atmospheric transmittance for each gas was computed using PcModWin6 (MODTRAN v6) and compared against measured spectrum. In this study, spectra are analyzed using non-linear least squares spectral fitting algorithm (GFIT) developed by the California Institute of Technology, U. S. A. The Total Carbon Column Observing Network (TCCON) identified standard spectral windows were selected for retrieving the GHGs over the study site. With the present retrieval scheme, precision of the FTIR achieved are 0.17 % to 0.52 % for $CO_2$, 0.30 % to 0.77 % for $CH_4$, 6.33 % to 8.92 % for CO and 0.41 % to 0.75 % for $N_2O$, respectively. Observed little to clear diurnal and seasonal variations in $XCO_2$, $XCH_4$, $XCO$ and $XN_2O$ respectively. In this work, we compared FTIR retrieved GHGs against Orbiting Carbon Observatory-2 (OCO-2) retrieved $XCO_2$ and Measurements of Pollution in the Troposphere (MOPITT) retrieved $XCO$. With the present retrievals, comparative study yields a mean relative bias between ground-based FTIR retrieved $XCO_2$ and $XCO$ are -1.50 % and 0.60 % against OCO-2 ($XCO_2$) and MOPITT ($XCO$) respectively. Pearson correlation coefficient (r) between FTIR retrieval and satellite retrievals are 0.80 ($XCO_2$, N=14 co-located) and 0.85 ($XCO$, N=18), respectively.



## 1. Introduction

Emission of greenhouse gases (GHGs) such as carbon dioxide ($CO_2$), methane ($CH_4$) and nitrous oxide ($N_2O$) have largely increased in the atmosphere since pre-industrial time due to anthropogenic activity (Stocker et al., 2013). $CO_2$ concentration is increasing in the atmosphere consistently since pre-industrial time and even it has crossed 400 ppm concentration in the Antarctic region as observed during 2016 (Mahesh et al., 2018). $CO_2$ and $CH_4$ are two most abundant anthropogenic GHGs next to water vapor ($H_2O$), contributing positive radiative forcing that results global warming (Myhre et al., 2013 IPCC, AR5 chapter 8). Since last decade, anthropogenic $CH_4$ concentration has drawn research community attention due to its consistent increase in the atmosphere and uncertainty of source/sink (Huang et al., 2015). National Oceanic and Atmospheric Administration (NOAA) long-term observations show ~8 ppb year$^{-1}$ annual $CH_4$ increase (https://www.esrl.noaa.gov/gmd/ccgg/trends_ch4/). Annual total global $CH_4$ emission of 500±50 Tg estimation was bounded by its abundance and life time in the atmosphere (Dlugokencky et al., 2011). Major sources of emissions in India include agriculture residue and waste constituent. However, their ratio of contribution to $CH_4$ emissions is remain uncertain (Kirschke et al., 2013). Due to long-lived and positive climate forcing, $N_2O$ is also a powerful GHG that contributes to global warming. Atmospheric $N_2O$ concentrations have increased globally by 20% since 1750 and observed steady increasing rate of $0.73 \pm 0.03$ ppb year$^{-1}$ over the last three decades (IPCC, AR5). Atmospheric carbon monoxide (CO) is one of the ozone precursor gases, which also acts as an important GHG due to its significant role in the OH radical production (Crutzen and Zimmermann, 1991). CO gas is mainly emitted due to incomplete combustion from urban/industrial fossil-fuel, biofuel use and biomass burning. Atmospheric CO is also a sever pollutant thus plays an important role in examining the effect on air quality due to local and transported sources.

To know comprehensive understanding about source and sinks of GHGs requires accurate measurements with adequate spatial coverage. Current knowledge on GHG effect on climate change is mostly supported by surface based observations and also model simulations (Petri et al., 2012). Besides surface observations, column ($X$) measurements are also more reliable towards GHGs representation over a region, observed from the satellite platform. Global measurements on column $CO_2$ are especially important to understand the sources and sinks in regional level (Yokota et al., 2009). Rayner and O'Brien (2001) have reported that satellite based column observations on $CO_2$ with an accuracy and precision of 1-2 ppm are potential to develop an understanding of surface fluxes. Thus, ground and space based remote sensing together has become a powerful tool to address the spatial and temporal variability of GHG. Satellite based total columnar GHGs observations started in 2002 with the retrieval of $CO_2$ and $CH_4$ from the Atmospheric Infrared Sounder (AIRS) on-board Aqua platform and additionally CO and $N_2O$ from the Scanning Imaging Absorption Spectrometer for Atmospheric Cartography (SCIAMACHY) instrument on-board ENVISAT (Aumann et al., 2005; Xiong et al., 2008; Wagner et al., 2008). A consistent record of space-borne measurements of $CO_2$ and $CH_4$ have been available starting in 2009 from the Thermal and Near-infrared Sensor for carbon Observation on the Greenhouse gases Observation SATellite (TANSO-GOSAT; Feng et al., 2017). More recently, the Orbiting Carbon Observatory-2 (OCO-2) launched in 2014 has enabled



dedicated atmospheric $CO_2$ measurements with the high precision to identify the sources and sinks of $CO_2$ on regional scale (Wunch et al., 2017). Long-term space based global columnar CO
measurements are available from the Measurements of Pollution in the Troposphere (MOPITT), which is aboard the satellite Terra (Buchholz et al., 2017) and measures $X$CO in thermal and near infrared regions.

Total column measurements on GHGs can be performed using ground-based remote sensing of Fourier Transform Infrared Spectrometry (FTIR) measurement and retrieval techniques. Two
popular networks namely The Total Carbon Column Observing Network (TCCON) and Network for the Detection of Atmospheric Composition Change (NDACC) are established with ground-based FTIR instruments (models: IFS125-HR and IFS-120HR), which provides highly accurate and precise $CO_2$, $CH_4$, CO, $N_2O$ and other species in NIR and MIR region (Washenfelder et al., 2006; Messerschmidt et al., 2011; Sussmann et al., 2011; Gavrilov et al., 2014; Wang et al.,
2014; Pollard et al., 2017; Wang et al., 2017). Using the moon as a radiative source, Notholt et al. (1993) first reported column density of $N_2O$ and $CH_4$ observations during polar nights in the Arctic. At Shadnagar, India site is equipped with IFS 125M model which is presently not recommended in TCCON network. However a few studies are able to achieve the required accuracy and precision of GHGs retrievals (Petri, 2012). Recent TCCON measurements have
shown that the precision of the resulting mole fractions is about 0.15% for $CO_2$ and 0.5% for CO (Toon et al., 2009; Messerschmidt et al., 2010; Wunch et al., 2010). Therefore highly precise and accurate column measurements are decisive for estimating source and sinks as well as validating the satellite based retrievals.

First ground-based FTIR preliminary column retrievals of $CO_2$ (Mahesh et al., 2016), $CH_4$ and
$N_2O$ (Mahesh et al., 2017) over the Shadnagar region of India were attempted with in-house developed line-by-line radiative transfer algorithm (LBLRTA), which lead to coarse precision and accuracy. Objective of the present study is to report improved dry column-averaged concentration of $CO_2$, $CH_4$, CO and $N_2O$ using nonlinear least squares spectral fitting algorithm (GFIT-2014) obtained from the JPL/California Institute of Technology (Caltech). In this study,
we retrieved $X$$CO_2$, $X$$CH_4$, $X$CO and $X$$N_2O$ in the TCCON identified standard retrieval spectral windows while aiming to meet global accuracy and precision. Present study also compared OCO-2 retrieved $X$$CO_2$ and MOPITT retrieved $X$CO against ground-based FTIR retrievals over the study region.

## 2. Data measurement

Solar spectra are obtained under cloud free environment using ground-based remote sensing FTIR instrument (Make: Bruker Optiks, IFS125M model) installed at Atmospheric Science Laboratory (ASL) of NRSC, Shadnagar, India. The FTIR spectrometer is coupled with a solar tracker to track the sun for observations during clear-sky conditions. Spectra are acquired at spectral resolution ($\Delta v$) of 0.01 cm$^{-1}$ and optical path distance (OPD) of 90 cm with varied SZA
during the study period to obtain the representative atmospheric signal over the study region. Range of SZA from 10:00 hours to 16:00 hours at 2 hours interval during the study period were ~ [22°-48°], [5°-40°], [32°-52°], [60°-75°] respectively. Accuracy of the species retrievals depends on SZA of the site (Wunch et al., 2015). The $X$$CO_2$, $X$$CH_4$, $X$CO and $X$$N_2O$ errors (with



SZA dependency) are <0.25 % (until SZA ~ 82°), <0.50 % (SZA above ~85°), <4 % (decreases with SZA) and ~ 1 % (independent of SZA). Details of the instrument are described in Mahesh et al. (2016). As shown in Figure 1, the IFS125M spectrometer is optimized for solar measurements in near-infrared (NIR) region using Indium Antimonide (InSb) detector and Calcium Fluoride ($CaF_2$) beam splitter covering spectral range from 1800 $cm^{-1}$ to 11000 $cm^{-1}$ (5.50 μm to 0.90 μm). Wang et al. (2017) reported the root mean square error (RMSE) of fitting residuals of

Indium Gallium Arsenide (InGaAs) spectra are small compared to those of InSb spectra. Precision of the InGaAs is about two times better than the InSb. In this study, we utilized 50 days clear sky solar data collected during January 2016 to May 2016. During clear sky days, instrument was operated from 10:00 hours to 16:00 hours local time at spectral sampling rate of 5 minutes. Although DC recorded interferogram minimize the effect of source brightness

fluctuation due to changes in the atmosphere (Keppel-Aleks, 2007), in the present study, IFS125M is not enabled to DC signal. However, each day observations with internal NIR source's amplitude of 20,000 and the position of interferogram assured high quality recording of solar spectra. This is a proxy to check the instrument alignment and direct checking by the cell measurements filled with known pressure and temperature. The stable instrument line shape is an

important parameter in ground-based FTIR data retrievals (Hase, 2012). The alignment of the IFS125M at ASL is checked with $N_2O$ gas cell measurement (Mahesh et al., 2017). At the ASL, a log book entry is maintained to note down the cloud condition and other environmental parameters including temperature and humidity inside the FTIR room for subsequent quality checking of the observed data.


## 3. Data analysis methodology

By utilizing the 50 day solar spectra, column averaged dry concentration of $CO_2$, $CH_4$, CO and $N_2O$ are retrieved using the nonlinear least squares spectral fitting algorithm, GFIT model (2014 release), developed at JPL/Caltech (Toon et al., 1992). GFIT combines nonlinear iteration and

nonlinear least squares fitting to minimize the RMSE (Yuan et al., 2015). The TCCON have adopted GFIT,and PROFIT models to retrieve the column averaged dry concentration of atmospheric species (Toon et al., 1992; Notholt et al., 1993; Hase et al., 2004; Wunch et al., 2011). Retrieval windows of respective species are listed in Table 1.

Geographical features of the study location (Longitude, Latitude and Altitude) along with

automatic weather station measured meteorological parameters are supplied to the GFIT forward model to obtain high precision retrievals (Tran et al., 2010). Pressure, temperature and humidity profiles from the National Centers for Environment Prediction (NCEP) were used, and the a priori profiles were obtained from the Whole Atmosphere Community Climate Model (WACCM). Atmosphere is represented by 70 pressure levels in the line-by-line GFIT

code. Absorption coefficients, which are pressure and temperature dependent calculated line-by-line for each absorbing species at respective spectral band. In this study, WACCM and NCEP generated near real time a priori profiles are iteratively scaled to compute the vertical column density (VCD) with optimum RMSE. Causative factors of retrieval uncertainties include model generated *a prior* profile information of the site and local meteorological parameters besides


instrument stability and solar intensity variation (Gribanov et al., 2014). Computation of VCD using a priori profiles is given below (Kuai et al., 2012).

$$\text{VCD (gas)} = \text{scaling factor (gas)} \times \int_{Z_S}^{Z_T} \text{a priori profile (gas)} n dz --- (1)$$

where n is the total number of molecules in the column, $Z_S$ and $Z_T$ represents altitude at surface and top of the atmosphere. In this study, VCD is integrated from sun tracker height (0.54 km above mean sea level height) to 70 km. Column averaged-dry concentration of GHG, known $X$
(gas) is computed using VCD (gas)

$$X \text{ (gas)} = \frac{\text{VCD (gas)}}{\text{total dry} - \text{air column}, X \text{ (air)}} --- (2)$$

VCD of oxygen ($O_2$) in the atmosphere is well known and stable. Hence, total dry-air column can be derived using relation between VCD ($O_2$) and column dry-$O_2$ abundance (20.95 %) in the atmosphere.

$$X \text{ (air)} = \frac{\text{VCD } (O_2)}{0.2095} --- (3)$$

Equation 2 & 3 yield column averaged-dry concentration of a species as shown below

$$X \text{ (gas)} = \frac{\text{VCD (gas)}}{\text{VCD } (O_2)} \times 0.2095 --- (4)$$

Dry column–averaged concentration, $X$ (gas) accounts the influence of water vapor and change in surface pressure. Compared to VCD, $X$ (gas) reduces the system error sources that effect target gas and $O_2$ (Washenfelder et al., 2006). $X$ (gas) is the final representation of columnar retrievals from the FTIR data.

## 4. Results and Discussion

Measurements of direct solar spectra are obtained using ground-based remote sensing FTIR spectrometer during January 2016 to May 2016. Central wavenumbers (cm$^{-1}$) used in GFIT for computing the VCD of GHGs ($CO_2$, $CH_4$, CO, $N_2O$) and $O_2$ are given in Table 1. The final VCD (GHGs) is the average of all respective spectral windows. Dry column-averaged concentrations of GHGs were computed using Equation 4. Hourly and daily VCD ($O_2$) during the study period
is shown in Figure 2, which was further used to compute $X$ (GHGs). In the present retrieval, mean $X$ (air) value for current measurement site is about 0.97±0.007, which is typically about 0.98 for TCCON measurements and exhibits a small diurnal variation

During the study period, VCD ($O_2$) along with GHGs are retrieved at 5 minutes interval and averaged to hourly for understanding of diurnal of total column observations. Retrieved hourly
and daily mean VCD ($O_2$) over the study area are $4.21\times10^{24} \pm 7.36\times10^{22}$ (molecules cm$^{-2}$), $4.21\times10^{24} \pm 8.46 \times10^{22}$ (molecules cm$^{-2}$), respectively.


### 4.1. Diurnal and seasonal variation of dry column-averaged GHGs concentration

During the analysis procedure, around 3 % cloud contaminated solar spectra are manually removed based on log book entry on cloud condition at the ASL. Figures 3, 4a show time series retrieved VCD and dry column-averaged concentration of $CO_2$, $CH_4$, CO and $N_2O$ over the Shadnagar station during the study period. Daily averaged VCD ($CO_2$, $CH_4$, CO and $N_2O$) are $8.26 \times 10^{21} \pm 7.90 \times 10^{19}$, $3.71 \times 10^{19} \pm 4.69 \times 10^{17}$, $2.60 \times 10^{18} \pm 1.63 \times 10^{17}$ and $6.55 \times 10^{18} \pm 1.46 \times 10^{17}$ molecules $cm^{-2}$, respectively.

The hourly mean and 5 min $XCO_2$ range from 405 ppmv to 412 ppmv with maximum deviation of 7 ppmv during the study period. As shown in Figure 4b, the diurnal variation of $XCO_2$, $XCH_4$, $XCO$ and $XN_2O$ are observed with small diurnal amplitude of ~2 ppmv, ~10 ppbv, ~5 ppbv and ~3 ppbv, respectively. VCD of CO and dry column-averaged concentration of CO exhibit relatively high seasonality compared to other GHGs. This variability could be due to its varied 205 dynamic sources and short life time (~2 months). CO is largely produced by incomplete combustion from biomass burning and fossil fuels besides its chemical production through $CH_4$ and volatile organic compounds (Buchholz et al., 2017). Temporally co-located diurnal amplitudes of $XCO_2$ and $XCH_4$ are in the order of surface level $CO_2$, $CH_4$ diurnal amplitudes over the Shadnagar (Sreenivas et al., 2016). Variation in the surface level $CO_2$, $CH_4$ 210 concentration measured at ASL, Shadnagar are mostly associated with the local dynamics such as boundary layer height, photosynthesis activity and anthropogenic sources (Sreenivas et al., 2016). Since the present study site is also surrounded by a few small scale industries, emissions from local anthropogenic sources could influence GHGs. Thus, high precision dry column-averaged concentrations play an important role to understand the regional source and sinks 215 besides surface based observations.

Monthly averaged dry column-averaged concentration of $CO_2$, $CH_4$, CO and $N_2O$ are shown in Figure 5. Observed ~3-5 ppmv seasonal amplitude in the $XCO_2$ during the winter (January-February) and pre-monsoon (March-May). Less $XCO_2$ was observed in winter compared to pre-monsoon, which could be due to less $CO_2$ assimilation by decreasing temperature and solar 220 radiation in winter (Gilmanov et al., 2004). The $XCH_4$ exhibits little to moderate seasonality from winter to pre-monsoon with maximum amplitude change of ~0.015 ppmv. Dry column-averaged CO concentration shows relatively large seasonality across the study period, with amplitude of variation about 29 ppbv. This variation may be associated with day to day emission of CO from fossil fuels, biomass burning and chemical production. The $XN_2O$ shows similar 225 seasonality like $XCH_4$ with little to moderate seasonal variation. 1 % maximum deviation in $XN_2O$ is observed during the study period. Details of the spectra and time acquisition along with monthly mean of VCD (GHGs) and X (GHGs) are shown in Table 2

### 4.2. Comparative analysis between FTIR and Satellite retrievals

Present study reports preliminary comparative analysis between ground-based FTIR retrieved $XCO_2$ and VCD (CO) against OCO-2 retrieved $XCO_2$ and MOPITT retrieved VCD (CO) during the study period over the study site as shown in figure 6. The OCO-2 is first Earth-orbiting polar,



sun-synchronous satellite of National Aeronautics and Space Administration (NASA) for measuring space-based high precision atmospheric global $CO_2$ (Wunch et al., 2017) with a 16-day revisit cycle and crossing equator ~1:35 PM local time. OCO-2 measures $O_2$ and $CO_2$ in NIR spectral bands centered at 0.765, 1.61 (strong), and 2.06 μm (weak). Nadir and glint mode $CO_2$ observations are utilized in the present study to compare against FTIR retrieved $XCO_2$.


MOPITT is a nadir sounding instrument aboard the Terra satellite with a ground resolution of 22×22 km. MOPITT sense IR radiation emitted from the surface and measures total columnar CO with a correlation radiometer at thermal IR-NIR of 2.30 μm. MOPITT crosses equator at ~10:30 AM and 10:30 PM local time (Zhou et al., 2016; Buchholz et al., 2017). Spatio-temporal co-located approach has implemented in the present study for the comparison of satellite retrievals against FTIR retrievals. Figure 6 shows FTIR retrieved $XCO_2$ and VCD (CO) against OCO-2 retrieved $XCO_2$ and MOPITT VCD of CO, respectively.




OCO-2 and MOPITT retrievals are spatially co-located with ground-based FTIR by selecting the satellite grid size of 2°×2° around the FTIR site. Satellite foot print of 2°×2° is chosen optimally due to lack of co-located points in finer grid size, which could be one of the reasons for observed deviations. With this approach, present study could co-locate 14 points for $XCO_2$ and 18 points for $XCO$ respectively during the stud period. Daily and monthly trends of $XCO_2$ and VCD (CO) retrieved from FTIR follows satellite retrievals. Statistical comparison shows Pearson correlation coefficient 'r' and coefficient of determination ($R^2$) between FTIR retrieved $XCO_2$ and OCO-2 retrieved $XCO_2$ are 0.79 and 0.63 respectively. Bias, root mean square deviation (RMSD) and scatter index (SI) for the $XCO_2$ comparison show 6.20 ppmv, 1.71 ppmv and 0.004, respectively. SI is a good parameter to describe the accuracy of an estimate. MOPITT retrieved VCD (CO) and FTIR retrieved VCD (CO) show very good agreement with less deviations. The 'r' and $R^2$ between FTIR retrieved VCD (CO) and MOPITT retrieved VCD (CO) are 0.85 and 0.73 respectively. Bias, RMSD and SI for the VCD (CO) show -1.70×$10^{16}$ molecules cm-$^2$ (-0.81 ppbv), 2.90×$10^{16}$ molecules cm-$^2$ (1.44 ppbv) and 0.005, respectively.

## 5. Conclusions



The IFS125M FTIR spectrometer is operated at ASL, Shadnagar, India since March 2014. During 2016, IFS125M FTIR spectrometer was augmented to NIR region with Calcium Fluoride ($CaF_2$) beam splitter and Indium antimonide (InSb) detector while aiming to meet the TCCON recommended accuracy and precision of GHGs retrievals with IFS 120HR spectra. TCCON has recommended precision of the retrievals (1 σ) are <0.25 %, <0.30 %, < 1.0 % and < 0.50 % for $XCO_2$, $XCH_4$, $XCO$ and $XN_2O$ respectively. Present study utilized 50 days clear sky solar spectra to retrieve $XCO_2$, $XCH_4$, $XCO$ and $XN_2O$ in the TCCON recommended spectral windows. Following are the salient observations and findings in the present study.


- VCD of $CO_2$, $CH_4$, CO, $N_2O$ and $O_2$ are retrieved using GFIT model. Retrievals of the present study include WACCM simulated a priori profiles of GHGs and NCEP re-analysis meteorological parameters in GFIT model during the study period.
- With the present retrieval mode, achieved precision are 0.17 % to 0.52 % for $XCO_2$, 0.30 % to 0.77 % for $XCH_4$, 6.33 % to 8.92 % for $XCO$ and 0.41 % to 0.75 % for $XN_2O$,



respectively. Associated uncertainty of retrieval may depends on the inputs given in the forward model and stability of the ILS of the FTIR and sensitivity of the InSb detector.

- Future retrievals will focus on improving the inputs to the radiative transfer model and ensuring the stability of the instrument with $N_2O$ gas cell.
- Present study observed less diurnal to clear seasonal variations in the dry column-averaged concentration of GHGs.
- Comparative analysis show MOPITT retrieved $XCO$ is in agreement with the FTIR
retrieved $XCO$. FTIR retrieved $XCO_2$ shows relatively large deviation (-1.51 % relative bias) from OCO-2 retrievals, which will be our future focus of work on improvements.

The present work is the continuation of our earlier wok (Mahesh et al., 2016; Mahesh et al., 2017) to achieve better retrievals accuracy of the GHGs based on FTIR observations. With the present study and improvements in future retrievals, ground-based columnar measurements will
be decisive for validating the satellite retrievals and also to estimate accurate source and sinks of the region.

## 6. Acknowledgement

Authors of this paper sincerely thank Shri. Santanu Chowdhary, Director, NRSC for his kind encouragement for this work. We also greatly acknowledge Dr. Paul Wennberg, Caltech for his
extended support for implementing this work with GFIT.

## Conflict of Interests

The authors declare that there is no conflict of interests.







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



## Tables

Table 1 TCCON recommended retrieval spectral windows

| Target gas | Central wavenumber (cm$^{-1}$) | Spectral width (cm$^{-1}$) |
|---|---|---|
| $CO_2$ | 6220.00 | 80.00 |
| | 6339.00 | 85.00 |
| | 5938.00 | 116.00 |
| $CH_4$ | 6002.00 | 11.10 |
| | 6076.00 | 138.00 |
| | 4395.00 | 43.40 |
| $N_2O$ | 4430.00 | 23.10 |
| | 4719.00 | 73.10 |
| CO | 4233.00 | 48.60 |
| | 4290.00 | 56.80 |
| $O_2$ | 7885.00 | 240.00 |

Table 2 Monthly mean $X$ (GHGs) and VCD (GHGs) during the study period

| Month | $CO_2$ $(\frac{ppmv}{mol\,cm^{-2} \times 10^{21}})$ | $CH_4$ $(\frac{ppmv}{mol\,cm^{-2} \times 10^{19}})$ | CO $(\frac{ppbv}{mol\,cm^{-2} \times 10^{18}})$ | $N_2O$ $(\frac{ppbv}{mol\,cm^{-2} \times 10^{18}})$ | No of days No of Spectra | Time (Hour) |
|---|---|---|---|---|---|---|
| Jan | $406.86 \pm 0.86$ $8.18 \pm 2.44 \times 10^{19}$ | $1.8474 \pm 0.0056$ $3.71 \pm 1.47 \times 10^{17}$ | $123.53 \pm 8.12$ $2.50 \pm 1.58 \times 10^{17}$ | $323.38 \pm 1.33$ $6.51 \pm 4.29 \times 10^{16}$ | $\frac{07}{267}$ | 10-16 |
| Feb | $408.33 \pm 0.73$ $8.20 \pm 1.71 \times 10^{19}$ | $1.8471 \pm 0.0085$ $3.71 \pm 1.57 \times 10^{17}$ | $130.04 \pm 11.88$ $2.61 \pm 2.33 \times 10^{17}$ | $323.47 \pm 2.15$ $6.50 \pm 4.78 \times 10^{16}$ | $\frac{15}{685}$ | 10-16 |
| Mar | $411.26 \pm 4.53$ $8.25 \pm 3.46 \times 10^{20}$ | $1.8504 \pm 0.0143$ $3.77 \pm 1.39 \times 10^{18}$ | $143.23 \pm 14.43$ $2.92 \pm 3.90 \times 10^{17}$ | $327.72 \pm 8.37$ $6.54 \pm 4.43 \times 10^{17}$ | $\frac{09}{236}$ | 10-16 |
| Apr | $411.98 \pm 2.17$ $8.26 \pm 6.79 \times 10^{19}$ | $1.8433 \pm 0.0085$ $3.70 \pm 4.12 \times 10^{17}$ | $136.19 \pm 10.55$ $2.74 \pm 2.38 \times 10^{17}$ | $324.55 \pm 2.44$ $6.51 \pm 6.22 \times 10^{16}$ | $\frac{08}{388}$ | 10-16 |
| May | $412.22 \pm 1.62$ $8.26 \pm 1.82 \times 10^{19}$ | $1.8349 \pm 0.0193$ $3.66 \pm 7.53 \times 10^{17}$ | $114.51 \pm 13.92$ $2.29 \pm 3.12 \times 10^{17}$ | $327.05 \pm 8.66$ $6.53 \pm 2.70 \times 10^{16}$ | $\frac{11}{424}$ | 10-16 |






# Figures

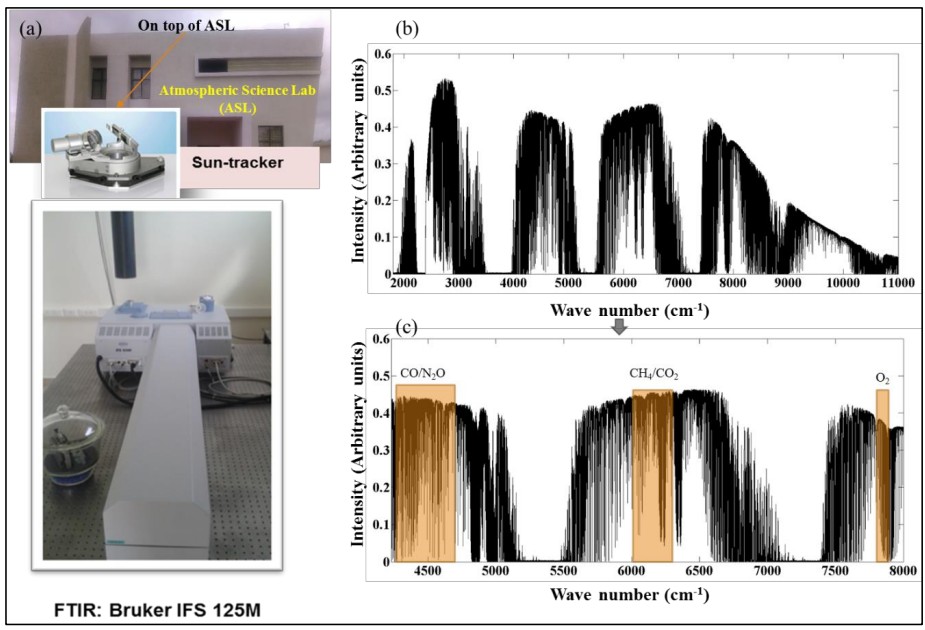

Figure 1 a) Atmospheric Science Lab with FTIR-IFS125M instrument, b) a single spectrum
recorded on 8$^{th}$ March 2016 and c) Retrieval windows for each gas

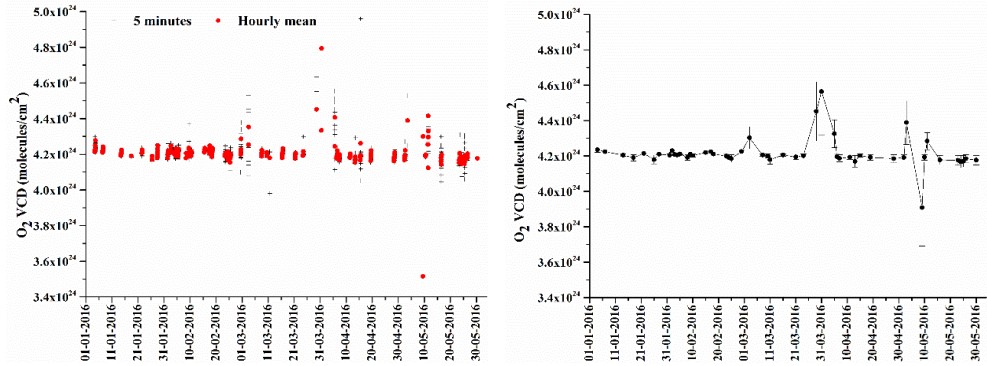

Figure 2 Retrieved VCD ($O_2$) a) 5 min and hourly mean b) Daily mean





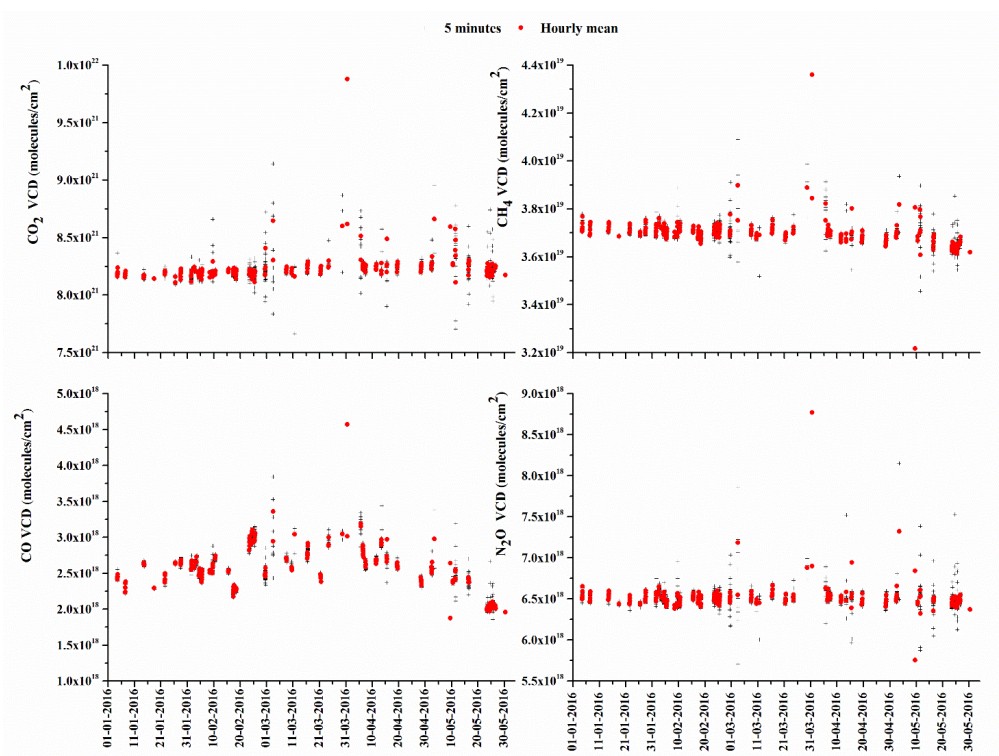


Figure 3 Red dots and black plus symbol indicate 5 minutes and hourly mean retrieved VCD ($CO_2$, $CH_4$, CO and $N_2O$) during the study period



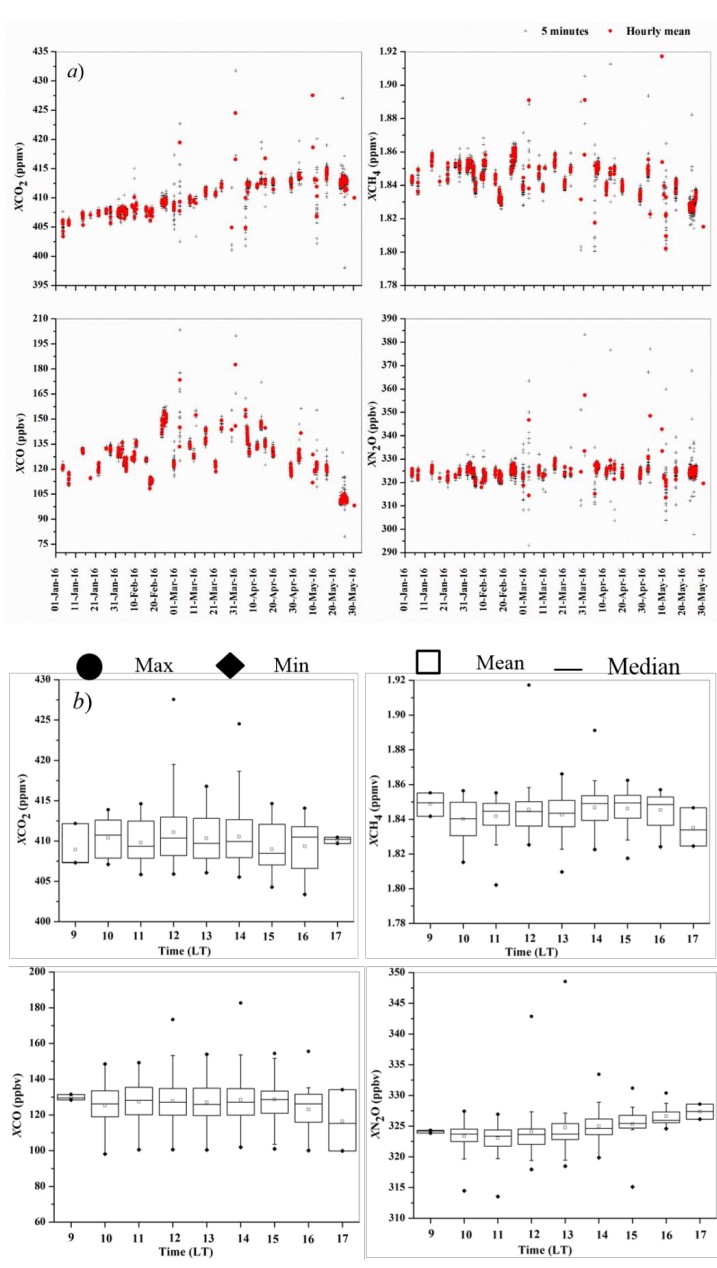

Figure 4 a) Red dots and black plus symbol indicate 5 minutes and hourly mean dry column-averaged concentration of $XCO_2$, $XCH_4$, $XCO$ and $XN_2O$ respectively during the study period; b) one day diurnal cycle of retrieved $XCO_2$, $XCH_4$, $XCO$ and $XN_2O$.






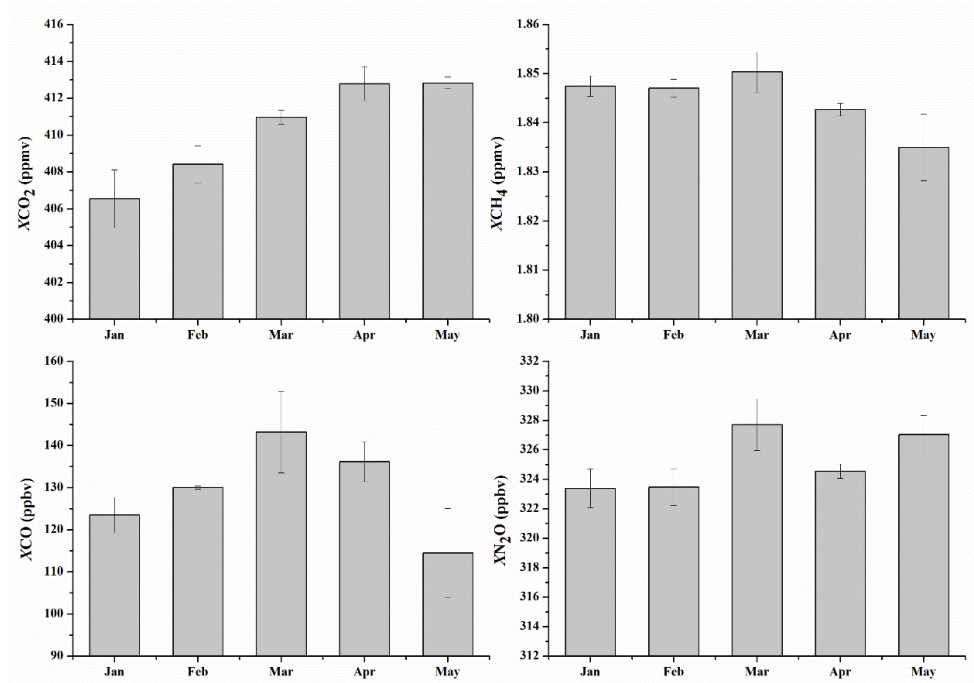

Figure 5 Monthly mean dry column-averaged concentrations of $CO_2$, $CH_4$, CO and $N_2O$







Figure 6 Validation of FTIR retrieved $XCO_2$ and $XCO$ against OCO-2 $XCO_2$ and MOPITT $XCO$ over the Shadnagar station, India.
