# Peer review of "Retrieval of CO2, CH4, CO and N2O using ground-based FTIR data and validation against satellite observations over the Shadnagar, India."

_Atmospheric Measurement Techniques, 2019_

## Referee Comment (RC1) · Anonymous Referee #1 · 4 Mar 2019

The authors present and discuss data of column-averaged greenhouse gas abundances collected during several months with a high-resolution FTIR spectrometer in Shadnagar, India. The information about a new ground-based spectrometer operated at this site is a valuable information and deserves publication, but the observation period is too limited and the elaboration presented in my opinion needs extension.

Why are only data recorded between start of and mid 2016 taken into account? Is the observatory inactive since?

Although GFIT is used for the data analysis, a huge bias in XCO2 versus OCO-II data is found (- 1.5%). I wonder whether the GGG suite and same retrieval setup is ap-

plied as used by TCCON? If not, the analysis of GHGs should be repeated with the standardized TCCON code.

A presentation of XAIR instead of the current figure showing VCD of O2 would be more useful for the reader for judging the level of stability achieved by the spectrometer. Presenting VCDs of other gases is also not too useful (XGas is the relevant quantity), so Fig. 3 could be removed.

I would suggest to include an intercomparison of the observed annual cycle with a model or climatological expectation for the variability of Xgas.

Detailed comments:

Abstract, line 31: not clear how the specified range of precision is established.

Introduction, line 45: consistently -> continuously

Introduction, line 55: should be: "their contribution to CH4 emissions remains uncertain"

Introduction, line 67: The Petri et al., 2012 reference is hardly appropriate here.

Introduction, line 73: offer the potential

Introduction, line 79: a record . . . has been available

Introduction, line 83: measurements with high precision

Introduction section starting line 88: it might be useful to mention the COCCON network in this context, see Frey et al., AMT, 2019 and references therein.

Section2, line 115: cloud free conditions

Section 2, line 116: omit make, Bruker Optics

Section 2, line 136: DC signal recording

Section 2: Please specify the data source or instrument of the ground pressure values

used for the data analysis

Section 3, line 151: The common code used by TCCON is GFIT, PROFFIT is used by several NDACC groups.

Section 3, line 156 ff: NDACC uses WACCM climatological profiles, the standardized GFIT TCCON analysis should not use this dataset.

Section 3, line 164: a priori

Section 3, line 177: omit "and O2", instead state "...that affect the target gases (Washenfelder et al., 2016)."

Section 4, line 188 ff: preferrably discuss XAIR, not VCD (O2), as the latter quantity varies with ground pressure.

---

## Referee Comment (RC2) · Anonymous Referee #2 · 2 May 2019

General comments:

This manuscript presents a study of greenhouse gases using a ground-based Fourier Transform Infrared Spectrometer of the type Bruker IFS 125M equipped with an InSb detector and CaF2 beamsplitter. The measured spectra are analyzed using the GFIT-2014 code and the retrieved VCD and Xgas products are presented. The XCO2 retrieved from the ground-based FTIR are compared to the XCO2 retrieved from OCO-2 and the XCO retrieved from the ground-based FTIR are compared to the XCO retrieved from MOPITT satellite observations.

The paper is poorly written and presented. Crucial information for the clarification of the

statements put forward is missing. One of my main criticisms is that the study covers a period of only 5 months with 50 days of measurements, which is very short time. As the measurement started approximately 3.5 years ago, the study period should be extended to at least a year. The other points are mentioned below in the specific comments section.

The authors highlight correctly that ground-based total column greenhouse measurement are very relevant in this part the world and will thus be an added value. However, these measurements need to be very precise and accurate to be useful for model studies or satellite validation. A proper demonstration over a longer period of time is therefore needed for the site. I recommend that the comments highlighted in this review should be addressed before it goes further in the review process and is considered for AMT publication.

Specific comments:

Page3 Line 122: What is meant by the range of SZA in the boxes? Are these examples from 4 days?

Page 4 Line 125: What kind of solar tracker is used for the measurement? This information is missing in this paper as well as in the reference paper of Mahesh et al., 2016. As this is the first demonstration of measurements it is important to give a description of the solar tracker and give a plot which shows the tracking accuracy of the solar tracker. This is relevant for the Xgas products.

As on each day observations were made with an internal NIR source, a plot of the ILS should be provided to show the stability of the instrument.

As mentioned in section 3. TCCON does not use the "PROFFIT" code for gas retrievals. PROFFIT (PROFile FIT) is a code used mostly by the NDACC-IRWG community.

What kind of a priori – daily?, monthly?, yearly? – is used from the WACCM and why? TCCON type retrievals use their own daily a priori generated from the TCCON a priori

generation tools. Those should be used for the analysis. Figure 2 should be exchanged with a plot of XAir and a zoom of the XAir for one long day of measurement should be shown. What is the reason for the variation of the O2 VCD in Fig 2 b?

Section 4.1: Discuss the results of Xgas values rather than the VCD of the gas products. As the VCD of gas contains some instrumental and measurement errors which are cancelled out while calculating the Xgas values.

The satellite comparison section is very weak. A detailed description should be given in relation to the satellite data - which version of the data is used, filter ... and what is the expected result for a similar co-incidence as selected in this paper. The reported bias is very high compared any other publications. This should be checked with either the same settings as other papers or using the settings of this paper for a few TCCON stations and compare the results to those of the satellite retrieved data. As it is now, the author makes several assumptions and nothing concrete is shown to prove them.

Page 7 line 263: Here I am totally confused, is IFS 120HR or IFS 125M being used for the study?

The authors do not show the measurement precision of the target gases. Rather they provide the upper and lower limits seen in the limited 5 month period. A clear demonstration of the measurement precision should be provided.

Page 8 line 274: the authors mentioned earlier that the ILS was very stable then why is it still in the error budget?

Please provide error bars in the top panel plots of Figure 6.

―――――――――――――――

---

## Author Comment (AC1) · 17 Jun 2019

Anonymous Referee #1 The authors present and discuss data of column-averaged greenhouse gas abundances collected during several months with a high-resolution FTIR spectrometer in Shadnagar, India. The information about a new ground-based spectrometer operated at this site is a valuable information and deserves publication, but the observation period is too limited and the elaboration presented in my opinion needs extension.

Reply to Referee#1 We appreciate your constructive comments. The comments and proposed corrections have been taken into account and helped to improve the pa-

per. Each comment has been addressed as follows. There is an extensive discussion among the authors regarding how to revise the content.

1.Why are only data recorded between start of and mid 2016 taken into account? Is the observatory inactive since? Reply: We sincerely appreciate your comments and suggestions. Present data were collected during clear sky days only. Data were not collected after May 2016 due to failure of HeNe source. We initially ordered FTIR 125M with MCT detector and KBr beam splitter configuration in 2014 and continued observations with this set up till 2015 during clear sky days. Later, we understand TCCON recommendations for precise column GHGs are different configuration. Hence immediately we started the procedure for augmenting the 125M system with InSb detector and CaF2 beam splitter while meeting TCCON standards. The IFS125M was augmented in December 2015 and started collecting NIR spectral data in 2016 only. Unfortunately HeNe laser source was failed in the middle of 2016. Hence we could not collect data beyond 2016 May. Therefore, the presented data analysis only focused on the available data in 2016. Instrument is again operational since March 2019. Future analysis will focus on long-term data analysis as suggested. Objective of the present study with the available data to attempt retrievals column GHGs using GFIT model while meeting the TCCON standards.

2.Although GFIT is used for the data analysis, a huge bias in XCO2 versus OCO-II data is found (- 1.5%). I wonder whether the GGG suite and same retrieval setup is applied as used by TCCON? If not, the analysis of GHGs should be repeated with the standardized TCCON code. Reply: We obtained GFIT (GGG, 2014V) and utilized the same retrieval setup as used by TCCON to process the data. Comparison approach is changed in the revised manuscript. We adopted median based method to compare OCO-2 retrieved XCO2 with FTIR retrieved XCO2 and found mean bias with RMSD is -2.82±3.01 ppm.

3.A presentation of XAIR instead of the current figure showing VCD of O2 would be more useful for the reader for judging the level of stability achieved by the spectrometer.

Presenting VCDs of other gases is also not too useful (XGas is the relevant quantity), so Fig. 3 could be removed Reply: Thanks for the comment. We have computed Xair for our site, which is shown in below figure and updated in the revised manuscript. The typical Xair value for TCCON measurements is about 0.98 and exhibits a small diurnal variation. The mean value of the current measurement site is 0.98 with a standard deviation of 0.006.

Figure. Xair for the measurement site. As suggested Figure 3 is removed from the revised manuscript. 4. I would suggest to include an intercomparison of the observed annual cycle with a model or climatological expectation for the variability of Xgas.

Reply: Please excuse us for not adding this result. Our division is not running GEOS-Chem model. However, we took the support to understand the GEOS-Chem column $CO_2$ simulation against FTIR retrieved $XCO_2$ during the study period. It is obserbed to be consistent in the trends with varied bias. Please see below figure for your reference.

Detailed comments:

Abstract, line 31: not clear how the specified range of precision is established. Reply: Precision for the current measurements are calculated on daily basis. In the present manuscript minimum and maximum achieved daily precision is reported. Introduction, line 45: consistently -> continuously Reply: updated in the manuscript Introduction, line 55: should be: "their contribution to $CH_4$ emissions remains uncertain" Reply: updated in the manuscript Introduction, line 67: The Petri et al., 2012 reference is hardly appropriate here. Reply: Thank you suggested reference, updated in the manuscript Introduction, line 73: offer the potential Reply: Updated as suggested. Introduction, line 79: a record : : : has been available Reply: updated in the manuscript Introduction, line 83: measurements with high precision Introduction section starting line 88: it might be useful to mention the COCCON network in this context, see Frey et al., AMT, 2019 and references therein. Reply: updated in the manuscript Section2, line 115: cloud free conditions Reply: updated in the manuscript Section 2, line 116:

omit make, Bruker Optics Reply: updated in the manuscript Section 2, line 136: DC signal recording Reply: updated in the manuscript Section 2: Please specify the data source or instrument of the ground pressure values used for the data analysis Reply: Dear Referee, ground pressure values are used from the Automatic Weather station data measured at the same location. Section 3, line 151: The common code used by TCCON is GFIT, PROFFIT is used by several NDACC groups. Reply: Thanks for the information. Updated in the manuscript Section 3, line 156 ff: NDACC uses WACCM climatological profiles, the standardized GFIT TCCON analysis should not use this dataset. Reply: The a priori profiles generated by the TCCON retrieval algorithm are based on the National Centre for Environment Prediction (NCEP) reanalysis data for temperature, pressure, and humidity. Section 3, line 164: a priori Reply: updated in the manuscript. Section 3, line 177: omit "and O2", instead state ": : :that affect the target gases (Washenfelder et al., 2016)." Reply: Thanks for suggestion. Updated in the manuscript Section 4, line 188 ff: preferrably discuss XAIR, not VCD (O2), as the latter quantity varies with ground pressure. Reply: Thanks for the suggestion. We updated as suggested.

Please also note the supplement to this comment:
https://www.atmos-meas-tech-discuss.net/amt-2019-7/amt-2019-7-AC1-supplement.pdf
* * *

---

## Author Comment (AC2) · 17 Jun 2019

General comments:

This manuscript presents a study of greenhouse gases using a ground-based Fourier Transform Infrared Spectrometer of the type Bruker IFS 125M equipped with an InSb detector and CaF2 beamsplitter. The measured spectra are analyzed using the GFIT-2014 code and the retrieved VCD and Xgas products are presented. The XCO2 retrieved from the ground-based FTIR are compared to the XCO2 retrieved from OCO-2

and the XCO retrieved from the ground-based FTIR are compared to the XCO retrieved from MOPITT satellite observations. Reply to Referee#2 We appreciate your constructive comments. The comments and proposed corrections have been taken into account and helped to improve the paper. Each comment has been addressed as follows.

The paper is poorly written and presented. Crucial information for the clarification of the statements put forward is missing. One of my main criticisms is that the study covers a period of only 5 months with 50 days of measurements, which is very short time. As the measurement started approximately 3.5 years ago, the study period should be extended to at least a year. The other points are mentioned below in the specific comments section.

The authors highlight correctly that ground-based total column greenhouse measurement are very relevant in this part the world and will thus be an added value. However, these measurements need to be very precise and accurate to be useful for model studies or satellite validation. A proper demonstration over a longer period of time is therefore needed for the site. I recommend that the comments highlighted in this review should be addressed before it goes further in the review process and is considered for AMT publication. Reply to Referee#2 We sincerely appreciate your comments and suggestions. Present data were collected during clear sky days only. Data were not collected after May 2016 due to failure of HeNe source. We initially ordered FTIR 125M with MCT detector and KBr beam splitter configuration in 2014 and continued observations with this set up till 2015 during clear sky days. Later, we understand TCCON recommendations for precise column GHGs are different configuration. Hence immediately we started the procedure for augmenting the 125M system with InSb detector and CaF2 beam splitter while meeting TCCON standards. The IFS125M was augmented in December 2015 and started collecting NIR spectral data in 2016 only. Unfortunately HeNe laser source was failed in the middle of 2016. Hence we could not collect data beyond 2016 May. Therefore, the presented data analysis only focused on the available data in 2016. Objective of the present study with the available data to attempt

[Figure]

retrievals column GHGs using GFIT model while meeting the TCCON standards.

Specific comments:

Page3 Line 122: What is meant by the range of SZA in the boxes? Are these examples from 4 days? Reply: SZA reported in the manuscript are calculated during the time period between 09:00 hr local time to 17:00 hr local time during the study period. The range of SZA [min SZA-max SZA] during the measurement period, i.e. during January 2016 to May 2016 are [5°-75°]. Page 4 Line 125: What kind of solar tracker is used for the measurement? This information is missing in this paper as well as in the reference paper of Mahesh et al., 2016. As this is the first demonstration of measurements it is important to give a description of the solar tracker and give a plot which shows the tracking accuracy of the solar tracker. This is relevant for the Xgas products. Reply:We have now provided the FTIR 125M measurement specifications along with sun tracker details in Table 1 in the revised manuscript.

Camtracker mode

As on each day observations were made with an internal NIR source, a plot of the ILS should be provided to show the stability of the instrument. Reply: Due to non-availability of gas cells we could not perform the ILS analysis regularly. However during 2015 and 2019 March, Service Engineer from Bruker optiks GmbH was tested the instrument stability with the N2O gas cell. With the support of one of the co-authors, ILS analysis performed (Figure 2). Details of the gas cell specifications are given in Table 1.

As mentioned in section 3. TCCON does not use the "PROFFIT" code for gas retrievals. PROFFIT (PROFile FIT) is a code used mostly by the NDACC-IRWG community. Reply: We have changed in the revised manuscript.

What kind of a priori – daily?, monthly?, yearly? – is used from the WACCM and why? TCCON type retrievals use their own daily a priori generated from the TCCON a priori

generation tools. Those should be used for the analysis. Figure 2 should be exchanged with a plot of XAir and a zoom of the XAir for one long day of measurement should be shown. What is the reason for the variation of the O2 VCD in Fig 2 b? Reply: We replace the sentence "Pressure, temperature and humidity profiles from the National Centers for Environment Prediction (NCEP) were used, and the a priori profiles were obtained from the Whole Atmosphere Community Climate Model (WACCM)" with "The a priori profiles generated by the TCCON retrieval algorithm are based on the National Centre for Environment Prediction (NCEP) reanalysis data for temperature, pressure, and humidity".

As suggested XAir is calculated for the analysis period and highlighted one day data as shown in figure 2 (a-b) in the revised manuscript.

Section 4.1: Discuss the results of Xgas values rather than the VCD of the gas products. As the VCD of gas contains some instrumental and measurement errors which are cancelled out while calculating the Xgas values. Reply: We have revised the manuscript as suggested.

The satellite comparison section is very weak. A detailed description should be given in relation to the satellite data - which version of the data is used, filter . . . and what is the expected result for a similar co-incidence as selected in this paper. The reported bias is very high compared any other publications. This should be checked with either the same settings as other papers or using the settings of this paper for a few TCCON stations and compare the results to those of the satellite retrieved data. As it is now, the author makes several assumptions and nothing concrete is shown to prove them. Reply: As suggested, we have added satellite data information in the revised manuscript. Comparative method has been changed in the revised manuscript. Below figure shows bias between OCO-2 retrived CO2 and TCCON sites data. Maximum mean bias of 5 ppm is shown in the below figure (sourced from Wunch et al. 2017)

The site-to-site differences between the OCO-2 data and the coincident TCCON data

are reported in recent study by Wunch et al. (2017). In the revised work, we also approached similar method and found mean bias with standard deviation is -2.82±3.01 ppm.

Page 7 line 263: Here I am totally confused, is IFS 120HR or IFS 125M being used for the study? Reply: We are using IFS125M spectrometer for NIR solar spectra collection. We have changed it to IFS125M in the manuscript.

The authors do not show the measurement precision of the target gases. Rather they provide the upper and lower limits seen in the limited 5 month period. A clear demonstration of the measurement precision should be provided. Reply: As suggested, total precision during the study period for respective gases given in the revised manuscript at Table 3.

Page 8 line 274: the authors mentioned earlier that the ILS was very stable then why is it still in the error budget? Reply: Due to non-availability of gas cells, we could not perform ILS analysis regularly. Results of ILS analysis during December 2015 data shown in figure 2 in the revised manuscript.

Please provide error bars in the top panel plots of Figure 6. Reply: Thanks for the comment. We have now updated the figure 6 as figure 7 in the revised manuscript.

Please also note the supplement to this comment:
https://www.atmos-meas-tech-discuss.net/amt-2019-7/amt-2019-7-AC2-supplement.pdf